# Closing the publishing gender gap in economics and political science: Does a critical mass matter?

**Daniel Stockemer**¹*, **Gabriela Galassi**², **Engi Abou-El-Kheir**¹

**1** Faculty of Social Sciences, University of Ottawa, Ottawa, Ontario, Canada, **2** Bank of Canada, Ottawa, Ontario, Canada

* daniel.stockemer@uottawa.ca

## Abstract

Using novel data on publications and citations by researchers in economics and political science in top 50 universities globally, we analyze the relationship between female representation and the gender gap in research output and impact. Using the concepts of substantive representation and critical mass, we expect that the average female researcher in departments with more women publishes more and receives more citations than the average female researcher in universities with a lower share of female faculty. Comparing women's publication and citation performance for the top 50 universities across the globe relative to men's, we find support for these expectations. We find that female researchers' performance matches that of their male counterparts in balanced departments, which are more common in political science. In contrast, the link between female performance and representation is weaker in departments with little gender balance, which are more typical in economics. These findings highlight the importance of reaching a critical mass of female representation to close gender gaps in research output and impact.

## Introduction

U.S. president Barack Obama famously said in 2013, "Empowering women isn't just the right thing to do—it's the smart thing to do." [1]. Equality of opportunities for women and men has long been a focal point for both practitioners and researchers. Despite progress, many fields in academia still lag, even in the more female-friendly social sciences [2,3].

In this article, we address the following question: Does increased female representation in academia help achieve a level playing field in research output and impact for both men and women? We focus on two of the most quantitative social sciences, economics and political science, to test whether different levels of female representation trigger varying gender gaps in publications and citations. To answer our research question, we start by measuring the gender gap at the individual level. That

**Data availability statement:** The data is posted on the Harvard Dataverse: Daniel Stockemer; Gabriela Galassi; Engi Abou-El-Kheir, 2025, "Replication data for "Closing the publishing gender gap in economics and political science: Does a critical mass matter?"", https://doi.org/10.7910/DVN/G6TJNG, Harvard Dataverse, V1, UNF:6:fovZpW4AuZ5ueouOxLxBmQ== [fileUNF]

**Funding:** Daniel Stockemer gratefully acknowledges funding through the Konrad Adenauer Research Chair. The funders had no role in study design, data collection and analysis, decision to publish, or preparation of the manuscript.

**Competing interests:** The authors have declared that no competing interests exist.

is, we compare the average female scholar to the average male scholar in terms of research output and impact. As such, our approach deviates from the general literature documenting the publication and citation gender gaps in both political science and economics. The existing literature primarily focuses on aggregate-level gender gaps, where most published content is written by male scholars [4–12]. We find similar results in terms of citations in political science [13–15] and some mixed evidence in economics [6,16–19].

As a starting point for our research, we posit that men's overrepresentation in publications and citations is at least in part an artifact of men's overrepresentation in the two disciplines of economics and political science [2,3]. Hence, the aggregate literature does not allow us to test whether there are differences in the individual male and female scholar's performance when it comes to publications and citations. In fact, there is only a very small literature assessing the individual gender gaps in academics' performance. In political science, several studies hint at such an individual publication gap by comparing the percentage of female authors in journals to their membership share in professional associations [20–22]. In economics, Ductor et al. [23] document an individual publication gap using data from EconLit (a database assembled by the American Economic Association), while other works focus on non-tenured faculty ([24] or for data before 2000s [25–27]).

We build on the scarce, but budding literature, which discusses men and women's publication performance to detect the existence and the magnitude of the gender gap in publishing. We deviate from this literature in one important aspect. Contrary to existing studies, we do not use one or several publications as the unit of analysis. Rather, we use the individual scholar as observational unit, and then aggregate this information for university departments. We believe that this approach provides more definitive evidence on the existence and the size of the gender gap in academic output in economics and political science. First, analyzing gender gaps at the individual level better aligns with productivity metrics. Second, aggregating at the department level captures the collaborative environments, in which men and women operate; thus, allowing for a fruitful comparison between disciplines; in our case political science and economics. In more detail, we constructed a novel dataset by web-scraping bibliographical information on all scholars in the economics and political science departments with a Google Scholar Page of the top 50 universities according to the QS Universities Ranking in 2023. Using first names to assign gender, we compare research output and citations of women and men. Our data comprises 1,715 scholars in economics (21 percent women) and 1,480 in political sciences (34 percent women). Our results confirm the existence of a gender gap at the individual level in both economics and in political science; the average woman publishes fewer papers and receives fewer citations than the average man.

More importantly, we explore one potential factor contributing to the gender gap: female representation in the profession. According to the concepts of substantive representation and critical mass [28–31] women in departments with a high share of female faculty members should outperform those in departments with fewer female faculty members in terms of publications and citations relative to men's. We also

believe that greater representation might foster stronger networks, and equitable access to resources, among others. Effectively, we find that women's output and influence relative to men's increases in more gender balanced departments. The relationship is more pronounced in political science—a field with comparatively high percentages of women faculty—compared to economics—a field with lower shares of female faculty members. Our findings suggest that higher female representation appears to decrease barriers that prevent women from publishing on par with men, lending support to the theories of substantive representation and critical mass.

The remainder of the paper continues as follows: first, we present the conceptual framework and formulate the hypotheses that guide our empirical analysis. Second, we provide further details on the data and the analysis we conduct. Next, we present the results of our analysis of how female representation relates to gender gaps in publications and citations. Finally, we conclude and provide some avenues for future research.

## Conceptual framework

In this study, we analyze the gender gap in academic publications and citations at the individual level, leveraging the general representation literature, which distinguishes between descriptive and substantive representation. Descriptive representation deals with the extent to which representatives stand for the represented, that is, the composition of institutions should mirror the composition of the represented [32]. In the context of our study, this means that we measure the numerical presence of female faculty in each department. Substantive representation, instead, refers to the congruence between representatives' actions and the interests of the represented [28,29]. Normally, studies discussing the link between the two concepts look at whether women represent women's interests in policy fields such as reproductive rights, family legislation, spending priorities or environmental legislation [33,34]. In our case, we evaluate whether women can publish and receive citations on par with men.

To assess whether women's descriptive representation influences their substantive representation in the academic world, we rely on the concept of critical mass. The idea behind critical mass theory is that a group of people, if it is big enough, and if it works together for the collective, can trigger collective action, which, if sustained, can become institutionalized [35,36]. Applied to women in politics, critical mass theory would predict that as more women are included/become present in certain spaces, they often work together or create alliances that influence and change the dominant (male) group, as well as possibly pave the way for more women to be represented in these spaces [37].

Adopting this logic in our context, we expect that a higher numerical presence of women in a department triggers a better performance by each female scholar in publications and citations; this relationship should further increase the more the level of female representation increases. That is, the theory of critical mass predicts not only a positive association between female representation and female-favorable outcomes, but also a more accentuated influence the more balanced the representation between men and women becomes. Many empirical studies find support for the prediction that increased numbers of women in politics should increase their chances to pass women-friendly policies increasingly favoring women [37–41]. In our context, multiple mechanisms may generate such an outcome: women in sizable numbers can push more women friendly policies, advocate gender equality and can set up mentoring schemes [10,42]; more female scholars in a department can also increase the collaboration opportunities for women who tend to work more often with other women [43–45]. All these features might potentially foster a more supportive environment that allows women to focus more effectively on research activities.

The literature has also discussed the group sizes that are necessary to set the processes behind critical mass theory in motion. For example, Kanter [30,31] distinguishes four types of groups: (i) uniform groups, composed only by men, (ii) skewed groups, dominated by men who control the culture of the group while women have little influence and are treated as tokens, (iii) tilted groups, still male-dominated but less skewed, allowing for some alliances to advance in the minorities' interests, and (v) balanced groups, where the minority status subsides and gender differences disappear as individual skills and abilities determine outcomes.

According to Kanter [30], skewed groups are particularly harmful for the advancements of women's interests. This is potentially because women in these groups often serve as tokens, who are not taken seriously by their male colleagues [30,31,46]. However, when we move from skewed to tilted groups, we expect that women receive better recognition for their published work and encouragement to fulfill their publication goals. Finally, when representation is balanced, we should move towards equality between men and women in the publication performance. More broadly, we can expect some synergies; more women in the discipline should increase the percentage of women in influential positions such as editors, editorial board members, and presidents in national and international professional associations. Female scholars should also be more numerous on grant applications and access better funding.

While critical mass is a powerful concept it is hard to operationalize. Although some literature has attempted to establish threshold numbers to move from one group type to another (e.g., [30] advances 20 percent as a threshold for the transition to tilted groups and 40 percent for the transition to balanced groups), we follow most of the literature (e.g., [47–49]) and expect these transitions to be fluid roughly around the benchmarks provided by Kanter [30].

From this conceptual discussion, we can formulate the following two hypotheses:

H(1): The individual publication and citation gap between men and women should be lower the more female professors there are per university department.

H(2): The effect of the female share in the department on limiting the individual publication and citation gap should be larger in political science—with a higher proportion of women—than in economics—where women are underrepresented.

## Methods

To test these hypotheses, we created a novel dataset comprising individual data on publication metrics for male and female economists and political scientists from the top 50 universities in the world, according to the QS University Ranking, 2023 version (see Table S1 for the list of universities included). We deliberately chose top universities because these institutions are research-oriented and produce a significant volume of research output. We excluded six Universities due to lack of identifiable departments in economics or political science, or missing faculty information. These are: Imperial College London, EPFL, PSL University, Korea Advanced Institute of Science and Technology, Institute Polytechnique de Paris, Tokyo Institute of Technology. We replace them with the six subsequent ones on the QS Ranking to maintain a total of 50. In economics, we also focused exclusively on economics departments and do not include business schools, as many have mandates beyond research, which could obscure the analysis of gender differences (e.g., [50] reported that women have achieved equality in promotion in academia, but only in institutions highly focused on research).

We further restricted the sample of scholars to those in full-time and permanent positions; in other words, we excluded postdoctoral fellows, visiting assistant professors and similar positions. Data collection took place between January and March 2024.

### Web-scraping faculty from departments' websites

As a first step, we web-scraped the names and positions of all faculty members from each economics and political science department. We successfully web-scraped data for 78 percent of economic departments and 66 percent of political science departments. We added the remaining 22 and 33 percent manually.

### Classification in gender and position

In the second step, we coded the gender of everyone according to their first name using the Gender API, an Application Programming Interface (API) that classifies names into gender (female and male) based on their frequency in

a name's dataset, combined with statistical analysis and machine learning. The Gender API first strips the first name from the professors' full names and uses these to classify them as male or female. Since gender is our key variable in the analysis, we manually verified the gender of all faculty members in Asian universities, those with names with three or fewer letters in other universities (as proxy for Asian-sounding names), and all others with an accuracy index below 80% according to the Gender API, as well as some randomly hand-picked cases. This manual check covered about 90 percent of the names and confirmed an overall classification accuracy of nearly 90 percent. While our coding is very thorough, it comes with the caveat that it might exclude non-binary identities and may lead to misgendering, as it does not rely on self-identification. Throughout the paper, we also use the terms "male" and "female" interchangeably with "man" and "woman".

To exclude individuals that were not permanent or full-time faculty, we only kept those classified as assistant professor, associate professor, and full professor, based on keyword searches within the position-related text extracted from the faculty webpages. Additionally, we created a dummy variable for professors whose job titles include special mentions such as "distinguished," "chair," "honorary," or that contains a personal name in the position title. For these classifications, we used natural language processing packages combined with manual validation.

## Scraping bibliographic information from the Google Scholar author pages

To obtain the bibliographic information for our scholars, we first collected the Google Scholar ID for everyone in our dataset. We did this by conducting an automatic Google search with the keywords "Google Scholar," the relevant discipline ("economics" or "political science"), the clean name (name after removing special characters), and the university name. In a random sample of 100 individuals, we found an error rate of 9 percent, primarily due to short names or surnames. To improve our ID search accuracy, we manually obtained IDs for all faculty members in Asian universities (where short names are more common) or for those with names consisting of three or fewer letters, whether in the first or last name. Additionally, we conducted a second pass on the missing IDs, this time omitting the discipline from the keyword search. These efforts significantly improved the ID retrieval process.

The final step involved scraping the Google Scholar author page information for all professors for whom we had successfully obtained a Google Scholar ID in the previous step. We used another API, SerpAPI, to perform this task. Altogether, we succeeded in retrieving the Google Scholar author page for 75 percent of all scholars in economics and 69 percent in political science.

An outlier check yielded the detection of six errors of misattributed Google Scholar profiles, which were manually corrected. We retrieved data for 1,715 scholars in economics across 50 departments, averaging 34 faculty per department split in 27 men and 7 women, on average, per department. For political science, our data included 1,480 scholars, averaging 30 people per department (i.e., 20 men and 10 women per department) (see Table S2).

The variables collected for each scholar include the i10-index, which measures the number of publications with at least ten citations, the total number of citations, the citations per publication calculated as citations divided by i10-index, and the number of citations for the most cited publication. Throughout the paper, we use *i10-index* and *number of publications* interchangeably. These data are publicly available in Harvard Metaverse Repositories [51].

We think that Google Scholar constitutes a good source for bibliographic information at the individual level. It is the largest and most comprehensive search engine displaying academic content [52], indexing a broad range of academic content such as journal articles, working papers, books and book chapters, conference proceedings, and policy reports, across all disciplines, from all countries, and in all languages. Once scholars have created a Google Scholar profile, they can choose to have all published content by said author automatically added to her/ his website, or they can opt for manual input. Even when manual input is selected, Google Scholar's algorithms automatically identify and suggest potential publications by the author, and notifies scholars if they enabled this option (for more information, please check [53–56]). While limitations exist, such as occasional errors in attribution, double counting of publications—which authors can

manually correct—and the inclusion of content without the highest quality control, these issues have been mitigated over time through improvements in automatic algorithms and publisher coverage.

### Statistical analysis

We first present some descriptive statistics underlying the existence of an individual publication and citation gap. In more detail, we present the averages for the publication and citation metrics by gender, for each discipline. For each indicator, we calculate the average of each indicator for female scholars in the department as a percentage of the average for male scholars. A figure of 50 implies that women publish half the number of publications/receive half the citations relative to men, and a figure of 100 means that the two genders publish and get cited on par. We present these inverse gender gaps for the totals, and over the past six years.

After establishing the existence of such gender gaps, we display the relationship between female representation and these gaps (the inverse, as previously defined) visually, in scatter plots. In every chart, we include data for both economics and political science, adding locally weighted scatterplot smoothing (Lowess) curves. These non-parametric fits capture non-linear relationships in the data without assuming a specific functional form.

Next, we estimate departmental-level regression models linking the proportion of women to the gender gaps in publications and citations (i.e., i10-index, total number of citations, citations per publication, and citations of the most cited publication). We include a dummy variable for political science departments and interact this dummy with the proportion of women. We add several control variables. For example, we control for the percentage of assistant professors in a department. Assistant professors are the category where women's representation is the highest among all academic [57]. However, scholars at the assistant professor level are also the category with potentially the least publications and citations of all ranks. We also control for the proportion of distinguished professors. Together, these two variables help capture the academic rank structure of the department, which may also influence the relationship between female representation and the gender gaps in publications and citations. Finally, we control for country-specific effects and add a dummy variable for all countries. We believe that in the US and the UK, for instance, the pressure to publish is probably highest, and hence we expect departments in these two countries to have a different pattern in publications and citations [58,59].

## Results

As a preliminary step, we first establish whether there is a publication and citation gap in economics and political science. Table 1 displays the descriptive statistics. In the top 50 universities globally, we find that male scholars publish roughly twice as much as female scholars and receive twice as many total citations and citations for their most cited publication. Smaller gaps exist in citations per publication. The magnitude of these gaps is relatively similar in both disciplines and has slightly declined in recent years (see the data for the last six years in the last row of Table 1). These statistics thus illustrate that the observed publication gap, where men are more likely to be authors than women, is not only an artifact of men's overrepresentation in the discipline; they also have a relevant individual component.

Having established the existence of a gender gap in both disciplines at the individual level, we now analyze the relationship between the percentage of female faculty members per department and the gender gap in publications/ citations, addressing our hypotheses, H(1) and H(2). We find support for both hypotheses.

Figs 1–4 display the proportion of women by department on the horizontal axis, and our measure of the inverse gender gap for each publication and citation indicator on the vertical axis. Blue dots correspond to economics departments, and green diamonds represent political science departments. We show similar charts using bibliometric data from the last six years in Figs S1–S3 (i.e., we show all outcomes in the category last six years except for citations of most cited publication which are not available for this timeframe in Google Scholar). The consistency of results over this more recent timeframe lends additional confidence in support of the positive relationship between declining gender gaps and more female scholars in a department.

**Table 1. Summary statistics (average values) and gender gaps (average for women divided by average for men) in research output and impact, overall and in the last six years.**

| | Economics | | | | Political science | | | |
|---|---|---|---|---|---|---|---|---|
| | i10-index | Citations | Average citations per publication | Most cited publication | i10-index | Citations | Average citations per publication | Most cited publication |
| *Overall* | | | | | | | | |
| Men | 48 | 11,198 | 148 | 1,853 | 40 | 6,004 | 103 | 1,070 |
| Women | 27 | 4,462 | 118 | 957 | 23 | 2,440 | 81 | 519 |
| **Percent female output (relative to male), overall** | **55%** | **40%** | **79%** | **52%** | **56%** | **41%** | **78%** | **48%** |
| Observations | 1,715 | 1,715 | 1,691 | 1,715 | 1,480 | 1,480 | 1,453 | 1,480 |
| *Last six years* | | | | | | | | |
| Men | 32 | 3,863 | 88 | – | 27 | 2,366 | 68 | – |
| Women | 20 | 1,997 | 82 | – | 16 | 1,109 | 57 | – |
| **Percent female output (relative to male), last six years** | **61%** | **52%** | **93%** | **–** | **60%** | **47%** | **85%** | **–** |
| Observations | 1,715 | 1,715 | 1,683 | | 1,480 | 1,480 | 1,448 | |

Citations for the most cited publication are only available for the entire publication history; a corresponding indicator for the past six years is not available. The number of observations for average citations per publication is smaller than for the remaining indicators due to scholars with 0 publications.

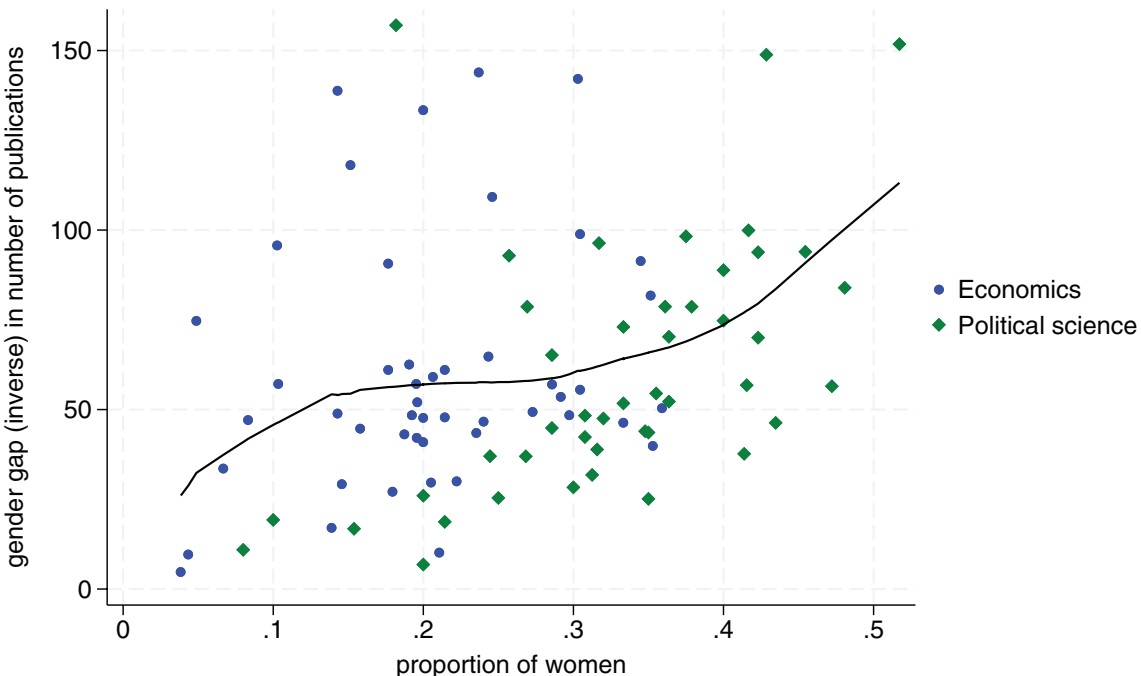

**Fig 1. Relationship between departmental proportion of women and women's average number of publications with at least ten citations (i-10 index) relative to men.** The x-axis shows the proportion of women in each department. The y-axis measures the inverse gender gap in the number of publications, defined as the average number of publications by female scholars as a percentage of the average number of publications by male scholars, by department. Blue dots correspond to economic departments and green diamonds to political science departments. The line represents a locally weighted scatterplot smoothing (lowess). Departments with 0 women in our dataset (2 in economics and 5 in political science) are excluded from the analysis.

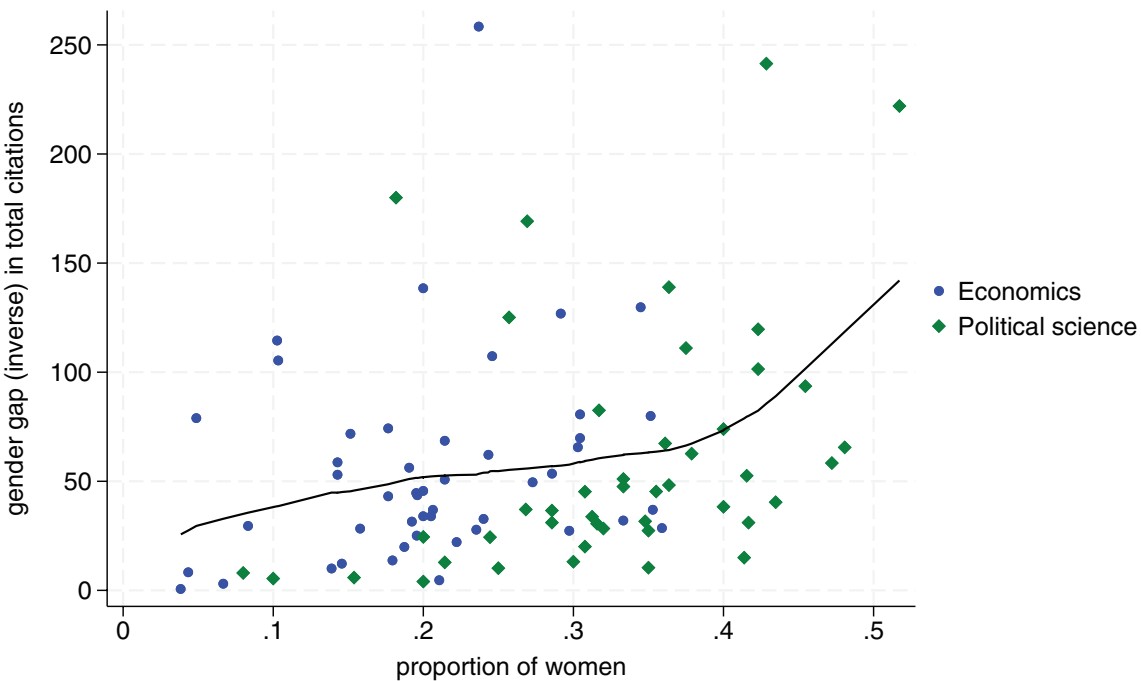

**Fig 2. Relationship between departmental proportion of women and women's average total citations relative to men.** The x-axis shows the proportion of women in each department. The y-axis measures the inverse of the gender gap in the total citations, defined as the average total citations by female scholars as a percentage of the average total citations by male scholars, by department. Blue dots correspond to economic departments and green diamonds to political science departments. The line represents a locally weighted scatterplot smoothing (lowess). Departments with 0 women in our dataset (2 in economics and 5 in political science) are excluded from the analysis.

In more detail, we see that the proportion of women per department is positively related to women's publication and citation performance for all publication and citation metrics. Overall, this implies that a higher proportion of women in a department triggers a better publication and citation performance by women relative to men. These effects are substantial. Fig 1 shows, for example, that in a political science department with 20 percent female members, a female scholar's publication output is roughly 40 percent that of a male scholar's output. In a department with 40 percent female members, female scholars publish at roughly 75 percent of the level of male scholars. Regarding citations, we see in Fig 2 that the gap closes from a ratio of roughly two to five in favor of men for a department with 20 percent women to a ratio of 7.5 to 10 for a department with 40 percent women.

Second, we find that the relationship between the proportion of women in departments and the reduction of the gender gap in publications and citations is stronger for political science, a field with generally more female faculty members, compared to economics, which has fewer female academics (i.e., 21 percent female faculty in economics versus 34 percent in political science in our sample). For all four dependent variables, the slope of the Lowess fit becomes steeper toward the right, which implies that the effect of more female scholars becomes stronger the more women are already in a department (see Figs 1–4).

The regression models confirm this observation with consistent point estimates, as shown in Table 2. The relationship between female representation and lower gender gaps is positive and significant across all metrics in political science and positive but significant only for total citations in economics. When controlling for department characteristics, no coefficient is significant in economics (even the point estimate for citations of the most cited publication becomes negative), and no qualitative changes are observed in political science. Substantively, there is also a difference in the effect of the

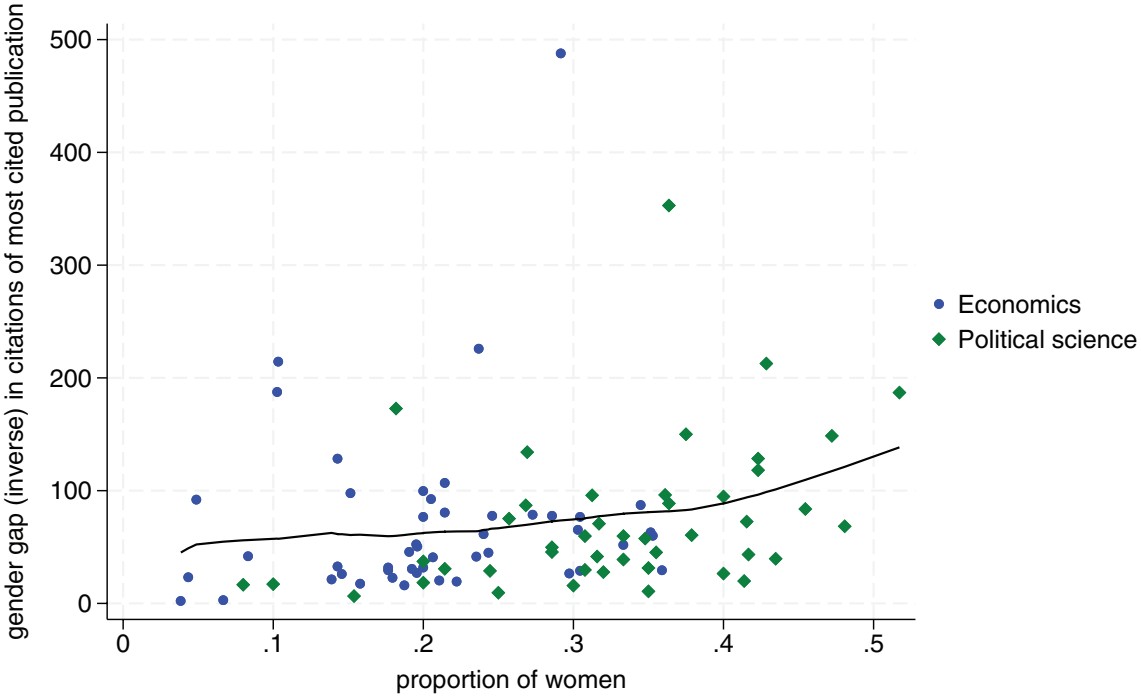

**Fig 3. Relationship between departmental proportion of women and women's average citations of most cited publication relative to men.** The x-axis shows the proportion of women in each department. The y-axis measures the inverse of the gender gap in the citations per publication, defined as the average citations per publication by female scholars as a percentage of the average citations per publication by male scholars, by department. Blue dots correspond to economic departments and green diamonds to political science departments. The line represents a locally weighted scatterplot smoothing (lowess). Departments with 0 women in our dataset (2 in economics and 5 in political science) are excluded from the analysis.

ratio between women and men scholars in departments, and the two genders (predicted) publication performance. For instance, the regression model with the i10-index as dependent variable and with controls predicts that for every 10-percentage point increase in the proportion of women, women's career output increases by 15 publications in political science and 3 publications in economics. For overall citations, a 10-percentage point increase in female faculty members triggers an estimated increase of 16 citations in political science compared to 14 citations in economics.

It is noteworthy from Figs 1–4 that no economics department has 40 percent or more women's representation. In contrast, in political science, there are several departments with a female share of professors between 40 percent and 50 percent. In accordance with the critical mass theory, these departments exhibit little difference in the publication performance between men and women. Most notably, the University of Melbourne and University of New South Wales, the two Australian universities in our sample, where women faculty represent 52 and 43 percent of faculty respectively, female scholars outperform male scholars on all publication and citations indicators. At the University of Cambridge, with 45 percent female faculty, women roughly perform on par with men, reaching on average 94 percent the number of average publications and citations than men.

Tying these observations to the previously discussed conceptual framework, it appears that many political science departments may have reached a tilted status, with some moving towards a balanced status. In contrast, economics departments still seem to vacillate between skewed and tilted status. Several economics departments have a minuscule share of female members, and no department in our sample has moved toward a balanced representation. This might also largely explain why there is still no single economics department in our sample where women perform on par with men both when it comes to publications and citations.

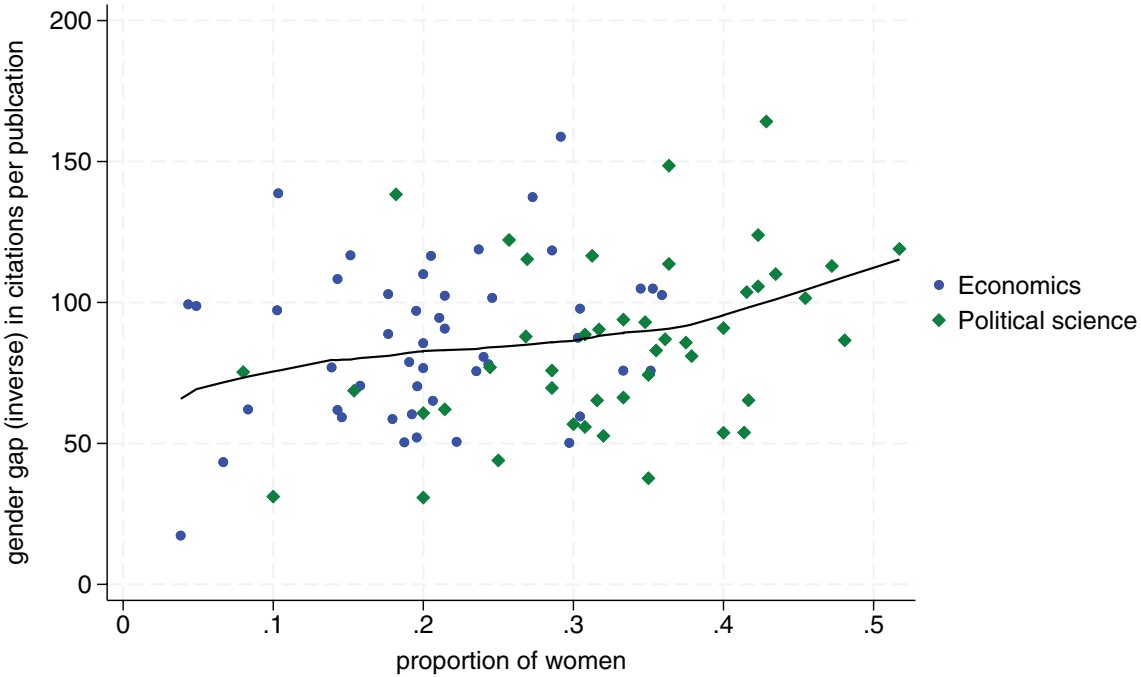

**Fig 4. Relationship between departmental proportion of women and women's average citations per publication relative to men.** The x-axis shows the proportion of women in each department. The y-axis measures the inverse of the gender gap in the citations per publication, defined as the average citations per publication by female scholars as a percentage of the average citations per publication by male scholars, by department. Blue dots correspond to economic departments and green diamonds to political science departments. The line represents a locally weighted scatterplot smoothing (lowess). Departments with 0 women in our dataset (2 in economics and 5 in political science) are excluded from the analysis.

**Table 2. Regression analysis of women's publication and citation averages relative to men's, on the proportion of women at the department level.**

| | Without controls | | | | With controls | | | |
|---|---|---|---|---|---|---|---|---|
| | DV i10-index | DV Citations | DV Most cited publication | DV Citations per publication | DV i10-index | DV Citations | DV Most cited publication | DV Citations per publication |
| Proportion of women | 88 (55.8) | 119* (69.8) | 112 (143.9) | 80 (54.1) | 31 (75.9) | 13 (97.4) | −60 (143.5) | 55 (69.4) |
| Political science | −42* (24.5) | −42 (34.6) | −54 (37.2) | −23 (19.8) | −43* (24.2) | -46 (33.4) | −77** (37.5) | −25 (19.5) |
| Proportion of women x Political science | 98 (81.3) | 100 (118.2) | 135 (164.7) | 39 (69.5) | 120 (86.0) | 149.7 (127.6) | 267 (165.5) | 52 (75.9) |
| Constant | 41*** (12.9) | 30** (15.0) | 47* (27.2) | 69*** (12.4) | 66** (26.4) | 93** (38.8) | 158** (62.4) | 94*** (23.2) |
| **Marg. Eff: Prop. of women Econ.** | **88 (55.8)** | **119* (69.8)** | **112 (143.9)** | **80 (54.1)** | **31 (75.9)** | **13 (97.3)** | **−60 (143.5)** | **55 (69.4)** |
| **Marg. Eff: Prop. of women Polit. Sc.** | **186*** (59.1)** | **220** (95.4)** | **248*** (80.2)** | **120*** (43.6)** | **151** (63.4)** | **163* (92.0)** | **206** (101.1)** | **107** (45.9)** |
| N | 93 | 93 | 93 | 93 | 93 | 93 | 93 | 93 |
| R² | 0.15 | 0.11 | 0.07 | 0.11 | 0.28 | 0.24 | 0.19 | 0.29 |
| Controls | no | no | no | no | yes | yes | yes | yes |

Robust standard errors are reported in parentheses. ***$p < 0.01$, **$p < 0.05$, *$p < 0.1$. Marginal effects for the proportion of women in economics (Econ.) and political science (Polit. Sc.) are derived from interaction terms. DV = dependent variable. Controls: country fixed effects, proportion of assistant professors and of distinguished professors.

## Conclusion

Using novel data on publications and citations by researchers in two prominent social sciences, namely economics and political science, we analyze the relationship between female representation and the research output and impact gender gap. After establishing the existence of these gaps, we show that a larger presence of women in departments helps reduce these gender gaps, especially when female representation is nearly balanced in the department. Our results provide strong support for the idea that promoting gender balance at the department level is crucial for the performance of female scholars. Women tend to publish more and receive more citations relative to men in departments where they are more than just tokens. In fact, in the few political science departments with parity or near parity among faculty, women perform on par with men. In contrast, in departments with few women, they publish considerably less than men and receive fewer citations.

As such our article provides evidence, even if this evidence is only indirect, that increasing the number of women can set in motion processes that favor the inclusion of women in the discipline. For example [36], affirms that achieving a critical mass of women can trigger what he labels 'interactive innovations' and 'reciprocal interdependence' (p. 142). The former concept refers to the idea that increasing the percentage of women in university departments can create progress in a non-linear fashion through sequential stages. The strong variation in publication output between men and women in various departments with similar percentage of female scholars shows that not everyone benefits from general progress in the same way. Yet, the discipline seems to have benefited as a whole, generating processes of reciprocal interdependence. These processes imply that more women in the discipline not only trigger women to publish more but should also increase the presence of women on editorial boards and as editors. This, in turn, could be a sign of a virtuous cycle of more empowerment. This interpretation comes with the caveat that there needs to be a critical mass of women to set these processes in motion. Due to the low percentage of female scholars in economics' departments, it seems that these processes have not started yet or only incompletely started there.

While we acknowledge that our findings may suffer from some limitations—such as the data being representative only of top research institutions in economics and political science, and the observational nature of the study limiting causal inferences—they suggest that the critical mass model is crucial for understanding the barriers to achieve gender parity in academia despite recent efforts to increase female participation. Women's publication and citation performance increases as departments move from skewed to tilted to balanced status. Among the top 50 world universities, no economics department has reached balanced status, generally lagging political science in terms of female representation. Consequently, it is not surprising that the impact of female representation on publication and citation performance is less pronounced in economics than in political science.

For policy, this article provides further evidence that increasing the number of women in academia works in that it can create a more equal playing field between the two sexes. This directly implies that hiring more women is essential, not only for creating gender-equal departments but also for enhancing women's academic performance. Until women achieve balanced representation, it is likely that men will continue to outperform women in academic productivity and impact.

## Supporting information

**S1 Table. List of universities included in the data.**
(DOCX)

**S2 Table. Summary statistics for the department level data.**
(DOCX)

**S1 Fig. Relationship between departmental proportion of women and women's average number of publications with at least ten citations (i-10 index) in the past six years relative to men.** Note: The x-axis shows the proportion of women in each department. The y-axis measures the inverse of the gender gap in the number of publications, defined as the average number of publications by female scholars as a percentage of the average number of publications by male

scholars in the past six years, by department. Blue dots correspond to economic departments and green diamonds to political science departments. The line represents a locally weighted scatterplot smoothing (lowess). Departments with 0 women in our dataset (2 in economics and 5 in political science) are excluded from the analysis.
(EPS)

**S2 Fig. Relationship between departmental proportion of women and women's average total citations in the past six years relative to men.** Note: The x-axis shows the proportion of women in each department. The y-axis measures the inverse of the gender gap in the total citations, defined as the average total citations by female scholars as a percentage of the average total citations by male scholars, by department. Blue dots correspond to economic departments and green diamonds to political science departments. The line represents a locally weighted scatterplot smoothing (lowess). Departments with 0 women in our dataset (2 in economics and 5 in political science) are excluded from the analysis.
(EPS)

**S3 Fig. Relationship between departmental proportion of women and women's average citations per publication in the past six years relative to men.** Note: The x-axis shows the proportion of women in each department. The y-axis measures the inverse of the gender gap in the citations per publication, defined as the average citations per publication by female scholars as a percentage of the average citations per publication by male scholars in the past six years, by department. Blue dots correspond to economic departments and green diamonds to political science departments. The line represents a locally weighted scatterplot smoothing (lowess). Departments with 0 women in our dataset (2 in economics and 5 in political science) are excluded from the analysis.
(EPS)

## Acknowledgments

This research has benefited from fruitful discussions with Reinhard Ellwanger, Nuno Paixao, and Shu Lin Wee. We are grateful to two anonymous referees for their fruitful comments and to Dave Campbell and Colleen Smith for their insights about data collection. The views expressed in this paper are those of the authors. No responsibility for them should be attributed to the Bank of Canada.

## Author contributions

**Conceptualization:** Daniel Stockemer.

**Data curation:** Gabriela Galassi, Engi Abou-El-Kheir.

**Funding acquisition:** Daniel Stockemer.

**Investigation:** Daniel Stockemer.

**Methodology:** Daniel Stockemer, Gabriela Galassi.

**Project administration:** Daniel Stockemer.

**Supervision:** Daniel Stockemer.

**Writing – original draft:** Daniel Stockemer, Gabriela Galassi.

**Writing – review & editing:** Daniel Stockemer, Gabriela Galassi.

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
