## [Decision Letter · Decision Letter 0]

6 Nov 2024

PONE-D-24-39523

Closing the publishing gender gap in economics and political science Does a critical mass matter?*

PLOS ONE

Dear Dr. Stockemer,

Thank you for submitting your manuscript to PLOS ONE. After careful consideration, we feel that it has merit but does not fully meet PLOS ONE’s publication criteria as it currently stands. Therefore, we invite you to submit a revised version of the manuscript that addresses the points raised during the review process.

I have received three reviews that provide detailed feedback on your manuscript. Two of the reviewers (Reviewer 1 and Reviewer 3) have provided insightful comments, raised important questions, and made valuable suggestions for improving the manuscript. As for Reviewer 2’s comments, I do not find them essential to the revision process, and you are not expected to respond to this review in your revision.

Based on the feedback provided, I am requesting a major revision of your manuscript. Please address each point raised by Reviewers 1 and 3 in detail. 

A rebuttal letter that responds to each point raised by the academic editor and reviewer(s). You should upload this letter as a separate file labeled 'Response to Reviewers'.A marked-up copy of your manuscript that highlights changes made to the original version. You should upload this as a separate file labeled 'Revised Manuscript with Track Changes'.An unmarked version of your revised paper without tracked changes. You should upload this as a separate file labeled 'Manuscript'.If applicable, we recommend that you deposit your laboratory protocols in protocols.io to enhance the reproducibility of your results. Protocols.io assigns your protocol its own identifier (DOI) so that it can be cited independently in the future. For instructions see: https://journals.plos.org/plosone/s/submission-guidelines#loc-laboratory-protocols . Additionally, PLOS ONE offers an option for publishing peer-reviewed Lab Protocol articles, which describe protocols hosted on protocols.io. Read more information on sharing protocols at https://plos.org/protocols?utm_medium=editorial-email&utm_source=authorletters&utm_campaign=protocols .

We look forward to receiving your revised manuscript.

Kind regards,

Andrey Lovakov, Ph.D.

Academic Editor

PLOS ONE

Journal Requirements:

2. Thank you for stating the following financial disclosure: “Konrad Adenauer Foundation, Canada Office”.

3. We note that your Data Availability Statement is currently as follows: “We collected the data ourselves. We will post it with the article as supplementary material.”

Please confirm at this time whether or not your submission contains all raw data required to replicate the results of your study. Authors must share the “minimal data set” for their submission. PLOS defines the minimal data set to consist of the data required to replicate all study findings reported in the article, as well as related metadata and methods (https://journals.plos.org/plosone/s/data-availability#loc-minimal-data-set-definition). For example, authors should submit the following data: - The values behind the means, standard deviations and other measures reported; - The values used to build graphs; - The points extracted from images for analysis. Authors do not need to submit their entire data set if only a portion of the data was used in the reported study. If your submission does not contain these data, please either upload them as Supporting Information files or deposit them to a stable, public repository and provide us with the relevant URLs, DOIs, or accession numbers. For a list of recommended repositories, please see https://journals.plos.org/plosone/s/recommended-repositories. If there are ethical or legal restrictions on sharing a de-identified data set, please explain them in detail (e.g., data contain potentially sensitive information, data are owned by a third-party organization, etc.) and who has imposed them (e.g., an ethics committee). Please also provide contact information for a data access committee, ethics committee, or other institutional body to which data requests may be sent. If data are owned by a third party, please indicate how others may request data access.

Reviewers' comments:

Reviewer's Responses to Questions

**Comments to the Author**

1. Is the manuscript technically sound, and do the data support the conclusions?

Reviewer #1: Partly

Reviewer #2: Partly

Reviewer #3: Yes

2. Has the statistical analysis been performed appropriately and rigorously? 

Reviewer #1: Yes

Reviewer #2: No

Reviewer #3: Yes

3. Have the authors made all data underlying the findings in their manuscript fully available?

Reviewer #1: No

Reviewer #2: No

Reviewer #3: No

4. Is the manuscript presented in an intelligible fashion and written in standard English?

Reviewer #1: Yes

Reviewer #2: Yes

Reviewer #3: Yes

5. Review Comments to the Author

Reviewer #1: This article analyzes the relationship between female representation and the gender gap in research output and impact. A distinctive feature of the study is that it assesses the gender gap in academia at the department level. The sample includes the Economics and Political Science departments of the top 50 universities worldwide, according to the QS University Ranking 2023. Information was collected from university websites about current scholars in full-time and permanent positions, excluding postdoctoral fellows, visiting assistant professors, etc. Subsequently, bibliometric data from Google Scholar was gathered for this sample. As a result, the study employs an extended list of publication types, including book chapters, monographs, and textbooks, thus attempting to overcome the limitations of the more common approach that uses only journal articles. Additionally, the publication overcomes the limitation associated with the difficulty of determining gender from Asian names through additional manual profile verification.

Several points are worth noting:

The approach used in the article, analyzing the gender gap at the department level, is not commonly seen in bibliometric studies due to its labor-intensive nature. As the authors rightly note, research more frequently focuses on discipline/university/country-level analyses. I would appreciate a more detailed explanation of the advantages of the department-level approach in the text.

The text provides a detailed and transparent description of the sampling and data collection process. In addition, more details on the features and limitations of the bibliometric database used would be beneficial. It is important to inform readers about how Google Scholar is formed, including the self-registration of accounts by researchers, how publications are added to a profile, etc. Addressing these points in the text could underscore the limitations of using Google Scholar as a source of bibliometric data for the purposes of this research.

The theoretical framework in the article is exciting, but a more detailed explanation of the mechanism underlying the critical mass theory as applied to the paper’s research question would be appreciated. Why might achieving a certain percentage of women on the faculty correlate with more publications and citations for women in the same department? For instance, one could presume that women might work more comfortably among women, potentially leading to a warmer working environment. Hence, resources that women might otherwise expend on additional communication in more masculine settings could be directed toward research activities. This guess is an example and does not necessarily explain the mechanism at work under critical mass theory in the context of gender gaps in academic publishing, but a more detailed description of the possible mechanisms in the theoretical framework section could be interesting.

On page 11, it is noted: “...when representation is balanced, we should not see any difference between men and women in publication performance,” which could be a strong statement within the research question posed. It is advisable for the article to mention other factors that could influence the gender gap in publications, even within the same department. For instance, years of work experience and the unequal impact of parenthood on mothers and fathers could be significant factors, considering the cumulative nature of used bibliometric indicators. Other important factors could include gender gaps in grant funding and career promotion, with explanations ranging from direct discrimination and gender stereotype reproduction to behavioral economics perspectives, such as a risk-averse etc. In any case, I believe a discussion of other factors that may influence the gender gap in publications and citations would be useful.

The article presents a list of universities included in the sample, but more information about the number of observations for each of the 50 universities would enhance comprehension. For example, expanding Table A1 to include "Total scholars" for each of the two disciplines and "% women.» Or it can be a new table with descriptive statistics on the total number of researchers and female researchers for the two disciplines (min, max, mean, SD, median) could be informative. I believe it's crucial for readers to understand the number of observations forming the assessments. If there are 1,718 researchers across 50 economics faculties, averaging 34 per faculty with 21% women, there would be an average of 7 women per faculty. However, universities may vary greatly in terms of overall numbers, and this distribution is important to see, especially when using average values in research. Let's suppose that, on average, there are 34 researchers per faculty; on average, 7 are women and 27 are men. The chance that among 27 men, there will be a researcher-«superstar» who will pull out the average value of bibliometric indicators for men of the whole faculty with his super-high indicators is higher than that such a «superstar» will be among 7 women. I'm not advocating for using median values instead, but readers must clearly comprehend how sensitive the faculty's average bibliometric indicators might be to adding or excluding a single observation, given the total number of observations.

The article uses two groups of bibliometric indicators from Google Scholar—overall indicators and the last six years indicators. The overall indicators reflect a person's career over its entire course. The linkage between overall career achievements and the current department can be quite ambiguous. Perhaps a researcher secured a professor position, let’s say, at Harvard after holding positions at departments with different gender balances and garnered main career achievements outside of Harvard. It’s quite challenging to affirm that the gender balance in this 2023 on Harvard faculty correlates with all cumulative publication achievements of a person. The second group of indicators—the last six years—offers a more accurate timeframe. Therefore, the gender balance in 2023 might be more convincingly related to a researcher’s output in this term. Perhaps viewing bibliometric indicators over even shorter periods could allow for establishing a firmer affiliation with the faculty. It's also worth noting potential concerns about the robustness of the percentage of women on small faculties. The current measure of women’s share in 2023 can be sensitive to hiring one additional person at a small faculty, significantly altering the gender balance.

Some minor clarifications:

On page 12, it's stated: “...we manually verified the gender of all faculty members in Asian universities.” Does this refer only to Asian universities, or were Asian names in universities from other countries, where such names can occur, also checked manually?

On pages 13-14, it's noted: “To control for country-specific effects, we add a dummy variable for the US and the UK, with all other countries serving as the reference category. In the US and the UK, the pressure to publish is probably highest, and hence we expect departments in these two countries to have a different pattern in publications and citations (Van Dalen 2021).” More detail is needed on why Australian and Canadian universities are grouped with Asian universities and why their institutional characteristics and publication patterns would be closer, for instance, to Japan than to the UK or US.

The term “article” is used several times in the text. Given that the database used comprises a broader list of publications than articles alone, a more general term such as "publications" might be suitable.

Reviewer #2: The article by Daniel Stockemer, Gabriela Galassi, and Engi Abou-El-Kheir represents a significant contribution to the literature on gender disparities in economics and political science. The work focuses on analyzing the impact of the proportion of women in university faculties on academic publication outcomes and citation rates. The central premise of the article is the "critical mass" theory, which suggests that achieving a certain level of female representation in academic departments can significantly reduce gender gaps in research output. Below, I first elaborate on the article’s strengths and then move on to a critical analysis of its limitations and weaknesses.

Strengths of the Article

1. Innovative Approach to Studying Gender Inequality

o The article addresses a highly relevant and important topic, namely gender inequality in academia, specifically within prestigious departments of economics and political science. The critical mass theory is a key element of the study and represents an innovative approach to the issue, which has not been widely applied in studies on gender disparities in academia. The authors aim to examine whether increasing the number of women in a department leads to improved publication outcomes for women and a reduction in the gender gap in citations and publications.

o The article goes beyond the traditional comparison of publication counts between men and women. The authors consider more complex metrics, such as citation counts and the number of articles with at least 10 citations (i10-index). This makes the study more comprehensive and offers a fuller picture of academic productivity.

o Importantly, the article analyzes two distinct fields—economics and political science—which allows for a better understanding of the differences in gender balance dynamics across different academic disciplines. The analysis of these two fields enables a comparison between academic environments with varying levels of female representation, adding an interdisciplinary dimension to the study.

2. Quality and Uniqueness of the Data Set

o One of the key strengths of the article is access to a unique data set, which was collected using modern techniques such as web-scraping and manual data verification. The authors created a data set covering the top 50 universities worldwide, enhancing the prestige and credibility of the findings.

o The data include publication counts, citation numbers, and female representation in political science and economics departments, allowing for a detailed analysis of the impact of gender on academic outcomes. Furthermore, through the use of tools like Gender API for verifying the gender of scholars based on their names, the authors ensured that the data were as accurate and reliable as possible, significantly improving the quality of the study.

o The high level of detail with which the authors approached data cleaning is another clear strength of the article. Not only did they ensure the correct gender assignment for researchers, but they also corrected inaccuracies in bibliometric sources concerning citation counts, further enhancing the reliability of the results. The authors performed manual corrections to eliminate erroneous data gathered from Google Scholar, which is a rare but commendable practice in many empirical studies.

3. Broad Institutional-Level Data Analysis

o The article focuses on analyzing the representation of women at the institutional level (departments), which is a valuable approach as it allows for an assessment of how structural characteristics of institutions affect academic outcomes. Instead of analyzing data at the individual researcher level, the authors opted for aggregated data for entire departments, providing insight into how institutions influence the variation in academic performance between men and women.

o Furthermore, this analysis allows for consideration of the specific characteristics of individual universities and their departments, enriching the interpretation of the findings. For instance, comparing prestigious institutions from different countries provides a better understanding of the global dynamics of gender disparities in academia.

4. Effort Invested in Data Cleaning

o Another strong point of the article is the considerable effort put into cleaning and verifying the data. The authors employed advanced methods to identify the gender of researchers and manually corrected data. For example, they manually checked and verified the gender of individuals with names that were not easily classified. They also manually eliminated errors related to citations in Google Scholar, which is a rare practice in many academic studies.

o Data cleaning is often overlooked but is a crucial element of empirical research, and the authors paid particular attention to this aspect, which enhances the quality of the analysis. The meticulous data cleaning reduces the risk of systematic errors that could affect the study's results.

5. Interdisciplinary Approach to the Issue

o The article takes an interdisciplinary approach by examining gender representation dynamics in two different fields: economics and political science. This diversity allows for a more comprehensive analysis, as it demonstrates that different disciplines have varying gender balance dynamics. Political science, with higher female representation (34% women), is contrasted with economics, which is more male-dominated (21% women), providing a better understanding of how different levels of female representation affect outcomes.

o Interdisciplinary comparison provides a broader perspective on gender disparities in academia and shows that solutions to gender balance problems may vary depending on the discipline. This approach opens up new perspectives for studying gender inequality without being confined to a single academic field.

6. Diverse Metrics for Measuring Academic Outcomes

o The authors used a wide range of metrics to assess academic productivity, including publication counts, citation numbers, citations per article, and the i10-index (number of articles cited at least 10 times). Incorporating these various metrics allows for a more comprehensive evaluation of academic productivity and helps avoid oversimplification that could arise from analyzing only publication counts.

o Using multiple metrics provides an opportunity to examine different aspects of academic success, from publication quantity to the quality and impact of the work (measured by citations). This gives the article a more detailed perspective on the productivity of men and women in academia.

7. Innovative Quantitative and Theoretical Approach

o The combination of a quantitative analysis with an original theoretical approach (critical mass theory) provides a strong scientific foundation for the article. The authors skillfully integrate empirical data with theory, making their study a valuable contribution to further research on gender disparities in the social sciences.

o The article does not limit itself to simple numerical comparisons—the authors also present theoretical frameworks and attempt to translate empirical findings into broader theoretical discussions, which further strengthens the article's value in the context of future academic research.

The article stands out for its innovative approach to studying gender disparities in academia, particularly through the application of critical mass theory, an interdisciplinary approach, and the high quality of its data. The authors placed great emphasis on the accuracy of the data they collected and cleaned, significantly enhancing the quality of the research. By including different disciplines and various metrics of academic success, the article offers a broad spectrum of analysis. Additionally, the use of critical mass theory introduces a new perspective into studies on gender equality in academia.

While the article makes an important contribution to the literature on gender disparities in economics and political science, it suffers from significant methodological and analytical limitations that hinder the depth and clarity of its findings. Below, I provide a detailed examination of the article’s weaknesses based on previous observations.

1. Limited Methodological Rigor: Simple Linear Regression Models

o The most significant weakness of the article lies in the use of simple linear regression models, which do not adequately capture the complexity of the phenomena being studied. The simple linear regressions employed by the authors are essentially little more than Pearson correlation analyses, examining the relationship between one independent variable (percentage of women in a department) and various dependent variables (e.g., publication counts, citations). This approach fails to account for the multi-dimensional nature of gender disparities in academia.

o Simple models ignore potential interactions between variables, which are likely crucial in understanding the complex relationship between women’s representation and academic outcomes. For instance, the impact of the proportion of women in a department might be moderated by other factors such as the institution's prestige, departmental policies, access to resources, or the size of the department. Without considering these interaction effects, the analysis remains too simplistic and overlooks important nuances.

o The lack of more advanced modeling techniques, such as multiple regression or models with interaction terms, significantly weakens the study’s explanatory power. Multiple regression would allow the authors to control for additional independent variables and provide a more comprehensive analysis of the various factors influencing publication outcomes. As it stands, the simple models employed in the article fall short of adequately capturing the complexity of the academic landscape.

2. Absence of Model Diagnostics

o The article does not include any information regarding model diagnostics, which is a major methodological shortcoming. In regression analysis, it is essential to test whether the model meets its fundamental assumptions (such as linearity, homoscedasticity, and lack of autocorrelation in residuals). Without such tests, it is impossible to assess whether the models used are correctly specified and whether the results can be trusted.

o The authors fail to conduct residual analysis or test for multicollinearity between independent variables. In multiple regression models, multicollinearity can greatly distort the interpretation of regression coefficients. The use of diagnostic tools, such as variance inflation factors (VIF), would allow the authors to evaluate whether collinearity issues are present, yet this crucial step is absent from the study.

o Moreover, tests for homoscedasticity (e.g., the Breusch-Pagan test) and autocorrelation of residuals (e.g., the Durbin-Watson test) are necessary to verify whether the models satisfy key regression assumptions. Without performing these diagnostic tests, the reliability of the models and the validity of the results remain uncertain. Ignoring these aspects calls into question the robustness of the conclusions drawn from the data.

3. Inadequate Treatment of Outliers

o Although the authors mention removing outliers from the data, they do not provide any detailed explanation of how these outliers were identified or how their removal affected the final results. The article lacks a clear description of whether outliers were identified based on univariate criteria (e.g., extreme values in a single variable) or multivariate criteria (e.g., unusual combinations of variables that deviate from the overall trend).

o Furthermore, the authors do not employ influence diagnostics—such as Cook’s Distance—to assess whether any individual observations had a disproportionate influence on the model estimates. Identifying influential outliers is a critical step in regression analysis, as outliers can skew results and lead to misleading conclusions.

o The absence of an analysis of model sensitivity—comparing results with and without the outliers—limits the reader's ability to evaluate how much the final models were impacted by these outlying data points. Without this analysis, it is unclear whether the removal of outliers improved the robustness of the models or whether it concealed important data trends.

4. Lack of Comprehensive Multivariate Analysis

o A major limitation of the article is the lack of multivariate analysis, which could have provided a more nuanced understanding of the factors influencing academic publication outcomes. The use of simple regression models, in which each outcome is analyzed in isolation, fails to account for the potential effects of other important variables (e.g., institutional prestige, availability of research funding, gender policies).

o A more thorough approach would have involved applying multiple regression techniques, which allow for the inclusion of several independent variables. This would enable the authors to control for confounding factors and offer a clearer picture of the true effect of the proportion of women in a department on academic success.

o Additionally, the authors do not consider interaction effects between variables, such as how the relationship between the proportion of women and publication outcomes might differ depending on the type of institution, available resources, or country-specific factors. Interaction terms in regression models would allow for a deeper exploration of these dynamics and could potentially reveal important moderating effects.

5. Low R² Values and Lack of Explanation

o Another significant weakness of the article is the presence of low R² values in the regression models, which indicate that the models explain only a small proportion of the variance in the dependent variables (publication counts, citations, etc.). Low R² values suggest that the models have limited explanatory power and fail to capture the full complexity of the relationships between the variables.

o The low R² values also raise concerns about the appropriateness of linear models. It is possible that the relationships between the proportion of women in a department and academic outcomes are nonlinear or influenced by unaccounted factors. As previously mentioned, more complex models, such as multiple regression or models with nonlinear terms, could potentially provide better fits to the data and explain more of the variance.

o Furthermore, low R² values may indicate that key variables are missing from the models. The authors could have considered including additional control variables, such as funding levels, departmental size, or institutional support for gender equality. These variables might have improved the models’ explanatory power and provided a clearer understanding of the factors influencing publication outcomes.

8. Generalization Issues: Limited Sample of Elite Institutions

o While the article focuses on data from the top 50 universities globally, this choice raises concerns about the generalizability of the findings. Elite institutions may not be representative of the broader academic landscape, and the dynamics of gender inequality in these prestigious institutions may differ significantly from those in less prestigious or smaller institutions.

o The study does not include an analysis of different types of institutions, such as research-focused universities vs. teaching-focused institutions, or public vs. private universities. These differences in institutional missions, funding structures, and academic priorities could have important implications for the gender disparities being studied, and their omission limits the scope of the article's conclusions.

o Additionally, the authors do not provide a comparative analysis across regions or countries, which could shed light on how different socio-political contexts influence gender disparities in academia. Including data from a more diverse range of institutions and geographic regions would have allowed for more comprehensive and widely applicable findings.

9. Lack of Advanced Models: Hierarchical or Path Analysis

o The absence of advanced statistical models, such as hierarchical (multilevel) models, is another significant limitation. Hierarchical models would have allowed the authors to account for the nested structure of the data, with researchers nested within departments and departments nested within universities. Such models are more appropriate for analyzing the effects of institutional and individual-level factors simultaneously and would have provided more robust estimates of the relationships between variables.

o Similarly, path analysis or structural equation modeling (SEM) could have been used to examine the complex causal relationships between the percentage of women, academic outcomes, and other contextual factors. These methods would have enabled the authors to model the direct and indirect effects of gender representation on academic success, offering a more nuanced understanding of the mechanisms at play.

The article’s main limitations stem from its inadequate use of statistical models, the absence of essential diagnostic tests, and the failure to account for the complexity of the relationships between variables. The reliance on simple linear regressions does not fully capture the multi-dimensional nature of gender disparities in academia. More sophisticated approaches, such as multiple regression, hierarchical models, or nonlinear modeling, could have significantly enhanced the explanatory power of the analysis. Additionally, low R² values, insufficient treatment of outliers, and the narrow focus on elite institutions limit the generalizability and robustness of the findings. To address these weaknesses, future research should employ more advanced statistical techniques and a broader range of institutions to provide a more comprehensive understanding of gender disparities in academia.

Reviewer #3: Summary:

Focusing on the fields of economics and political science, this article provides a comparative analysis of the relationship between women’s representation in science and gender gaps in research output and impact. The authors build a valuable manually curated dataset that will undoubtedly inform further research. They make a significant contribution to the field, as their work provides new evidence to understand the mechanisms shaping gender differences in scientific output. However, the presentation of the results should be improved.

I believe the article should be published after addressing minor revisions.

Main comments:

The authors should revise their Methods section in order to provide an accurate report of their research. For instance, the authors state that they “estimate bivariate and multiple regression models linking the proportion of women to the gender gaps in publications and citations (i.e., i10-index, total number of citations, citations per article, and citations of the most cited article)” (p. 13) but only bivariate analyses are presented in the article.

The distribution of the observations as depicted in Figures 1-4 and the R2 values reported suggest that a linear relationship is not the best functional form to model the relation between the variables. The authors should therefore provide a justification for their adoption of this model or explore the results of alternative modeling techniques.

The authors' justification for the inclusion of a dummy variable only in the cases of US and UK is not clear. They argue that in these countries “the pressure to publish is probably highest” (p. 14) and cite an article addressing the perceived work pressure in Dutch Universities. They should improve their arguments or include dummy variables for all countries.

The authors keep departments with no women in their analysis, assigning 0 to the inverse gender gap in these cases. The justification for the imputation of this value is not clear, and I would suggest removing these cases from the analysis or replacing them with other departments (namely, the departments in the next universities in the rank), as they are qualitatively different from the rest of the cases and may be introducing distortions.

Other comments:

I would suggest including Table A2 in the Main text, as it provides relevant contextual information. Moreover, information in this table is currently presented in an inconsistent manner: authors should include absolute values for Men and Women both for overall and for the last six years and include all variables in both cases (or justify their absence).

The authors infer gender relying on algorithmic and manual analysis of researchers’ names. This methodology excludes non-binary identities and may lead to misgendering, as it does not rely on self-identification. Authors should state this limitation. Furthermore, authors should revise their use of the terms “female”/“male” and “women” ”/“men” and adopt a consistent criterion.

The relationship between substantive representation—“the congruence between representatives’ actions and the interests of the represented”—and “assessing whether women can publish and receive citations on par with men” (p. 10) is not self-evident, but the authors fail to clearly explain it.

While the values for the Betas are reported with one decimal in Figures 1-4, they are rounded in Table 1, authors should use the same number of decimal places in both cases.

The authors state that all data underlying the findings described in their manuscript will be fully available without restriction, but at the moment of this review it is not yet available.

6. PLOS authors have the option to publish the peer review history of their article (what does this mean? ). If published, this will include your full peer review and any attached files.

**Do you want your identity to be public for this peer review?** For information about this choice, including consent withdrawal, please see our Privacy Policy .

Reviewer #1: No

Reviewer #2: No

Reviewer #3: No

---

## [Author Response · Author response to Decision Letter 1]

23 Jan 2025

Dear Dr. Lovakov,

Thank you very much for allowing us to revise our manuscript thoroughly. We also want to thank the three reviewers for their helpful comments. As per your instructions, I have been very thorough to address reviewer comments 1 and 2, but not 3. Please find below an explanation how we incorporated their comments.

Editor

Comment Response

All data needs to be publicly available We have added the dataset, codebook, and Stata code with this submission.

Reviewer 1

Comment Response

The approach used in the article, analyzing the gender gap at the department level, is not commonly seen in bibliometric studies due to its labor-intensive nature. As the authors rightly note, research more frequently focuses on discipline/university/country-level analyses. I would appreciate a more detailed explanation of the advantages of the department-level approach in the text We added a more detailed description on pages 2 and 3:

Among others we write:

We build on the scarce, but budding literature, which discusses men and women’s publication performance to detect the existence and the magnitude of the gender gap in publishing. We deviate from this literature in one important aspect. Contrary to existing studies, we do not use one or several publications as the unit of analysis. Rather, we use the individual scholar as observational unit, and then aggregate this information for university departments. We believe that this approach provides more definitive evidence on the existence and the size of the gender gap in academic output in economics and political science. First, analyzing gender gaps at the individual level better aligns with productivity metrics. Second, aggregating at the department level captures the collaborative environments, allowing for a fruitful comparison between disciplines; in our case political science and economics. In more detail, we constructed a novel dataset by web-scraping bibliographical information on all scholars in the economics and political science departments with a Google Scholar Page of the top 50 universities according to the QS Universities Ranking in 2023. Using first names to assign gender, we compare research output and citations of women and men. Our data comprises 1,718 scholars in economics (21 percent women) and 1,482 in political sciences (34 percent women). We establish a gender gap at the individual level in both economics and in political science, the average woman publishes fewer papers and receives fewer citations than the average man.

The text provides a detailed and transparent description of the sampling and data collection process. In addition, more details on the features and limitations of the bibliometric database used would be beneficial. It is important to inform readers about how Google Scholar is formed, including the self-registration of accounts by researchers, how publications are added to a profile, etc. Addressing these points in the text could underscore the limitations of using Google Scholar as a source of bibliometric data for the purposes of this research. We list on page 12 that the percentage of authors with Google Scholar information in our dataset is 75% in economics and 69% in political science.

We also added some information on page 13 to explain how Google Scholar performs the indexing. We write:

Once scholars have created their Google Scholar page, all published content by said author is automatically added to her/ his website (for more information, please check Scholastica 2024, and Harzing 2013, Halevi et al. 2017, and Delgado López-Cózar et al. 2019). While limitations exist, such as occasional errors in attribution, double counting of publications—which authors can manually correct—and the inclusion of content without the highest quality control, these issues have been mitigated over time through improvements in automatic algorithms and publisher coverage.

The theoretical framework in the article is exciting, but a more detailed explanation of the mechanism underlying the critical mass theory as applied to the paper’s research question would be appreciated. Why might achieving a certain percentage of women on the faculty correlate with more publications and citations for women in the same department? For instance, one could presume that women might work more comfortably among women, potentially leading to a warmer working environment. Hence, resources that women might otherwise expend on additional communication in more masculine settings could be directed toward research activities. This guess is an example and does not necessarily explain the mechanism at work under critical mass theory in the context of gender gaps in academic publishing, but a more detailed description of the possible mechanisms in the theoretical framework section could be interesting. We have addressed this point in several parts of the manuscript.

First, in the introduction, we added a clarification of the mechanism:

According to the concepts of substantive representation and critical mass (Pitkin 1967; Mansbridge 1999; Kanter 1977a,b) women in departments with a high share of female faculty should outperform those in departments with fewer female faculty in terms of publications and citations relative to men’s, as greater representation fosters stronger networks, equitable access to resources, among others.

Second, added two overview paragraphs in the conceptual framework section, where we explain the concept of critical mass more clearly and delve into the mechanisms (last sentence of the second paragraph). For example, on page 3 we write:

To assess whether women’s descriptive representation influences their substantive representation in the publication world, we rely on the concept of critical mass. The idea behind critical mass theory is that a group of people, if it is big enough, and if it works together for the collective, can trigger sustained collective action, which, if sustained, can become institutionalized (see Oliver et al. 1985; Grunenbaum 2015). Applied to women in politics, critical mass theory would predict that as more women are included/become present in certain spaces, they often work together or create alliances that influence and change the dominant (male) group, as well as possibly pave the way for more women to be represented in these spaces (Childs & Krook, 2008)

Adopting this logic in our context, we expect that a higher numerical presence of women in a department triggers a better performance by each female scholar in publications and citations, more when the level of female representation is higher. That is, the theory of critical mass predicts not only a positive association between female representation and female-favorable outcomes, but also a more accentuated influence the more balanced the representation between men and women becomes. Many empirical studies find support for the prediction that increased numbers of women in politics should increase their chances to pass women-friendly policies increasingly favoring women. (e.g., Studlar & McAllister 2002; Childs and Krook 2009; Kanthak & Krause 2012; Fokum et al. 2020; Mechkova & Carlitz 2021). In our context, multiple mechanisms may generate such an outcome: women tend to favor women without compromising quality (Bransch & Kvasnicka, 2022; Chari & Goldsmith-Pinkham, 2017) and are more likely to collaborate with other women (Ferber & Teiman, 1980; Piscopo et al., 2023; Ductor & Prummer, 2024), potentially fostering a more supportive environment that allows women to focus more effectively on research activities.

Finally, we speculated about some of the mechanisms that might be in play in explaining why a critical mass of women might trigger some higher publication performance in the conclusion.

As such our article provides evidence, even if this evidence is only indirect, that increasing the number of women can set in motion processes that favor the inclusion of women in the discipline. For example, Gruenbaum (2015) affirms that achieving a critical mass of women can trigger what he labels ‘interactive innovations’ and ‘reciprocal interdependence’ (p. 142). The former concept refers to the idea that increasing the percentage of women in university departments can create progress in a non-linear fashion through sequential stages. The strong variation in publication output between men and women in various departments with similar percentage of female scholars shows that noy everyone benefits from general progress in the same way. Yet, the discipline seems to have benefited as a whole generating processes of reciprocal interdependence. These processes imply that more women in the discipline not only trigger women to publish more but should also increase the presence of women on editorial boards and as editors. This, in turn, could be a sign of a virtuous cycle of more empowerment. This interpretation comes with the caveat that there needs to be a critical mass of women to set these processes in motion. Due to the low percentage of female scholars in economics’ departments, it seems that these processes have not started or only incompletely started there.

On page 11, it is noted: “...when representation is balanced, we should not see any difference between men and women in publication performance,” which could be a strong statement within the research question posed. It is advisable for the article to mention other factors that could influence the gender gap in publications, even within the same department. For instance, years of work experience and the unequal impact of parenthood on mothers and fathers could be significant factors, considering the cumulative nature of used bibliometric indicators. Other important factors could include gender gaps in grant funding and career promotion, with explanations ranging from direct discrimination and gender stereotype reproduction to behavioral economics perspectives, such as a risk-averse etc. In any case, I believe a discussion of other factors that may influence the gender gap in publications and citations would be useful. We agree that various factors that exceed female representation can influence gender gaps in academic output. We qualified the phrase alluded by the referee as

… when representation is balanced, we should move towards equality between men and women in the publication performance—. More broadly, we can expect some synergies; more women in the discipline should increase the percentage of women in influential positions such as editors, editorial board members, and presidents in national and international professional associations. Female scholars should also be more numerous on grant applications and access better funding.

We did mention other factors on page 7 and 8, We actually believe that the processes set in motion by a critical mass should benefit women in academia beyond publications and help them overcome structural barriers which will end up to level the playing in the future. For example, on page 5 we write:

More broadly, we can expect some synergies; more women in the discipline should increase the percentage of women in influential positions such as editors, editorial board members, and presidents in national and international professional associations. Female scholars should also be more numerous on grant applications and access better funding.

The article presents a list of universities included in the sample, but more information about the number of observations for each of the 50 universities would enhance comprehension. For example, expanding Table A1 to include "Total scholars" for each of the two disciplines and "% women.» Or it can be a new table with descriptive statistics on the total number of researchers and female researchers for the two disciplines (min, max, mean, SD, median) could be informative. I believe it's crucial for readers to understand the number of observations forming the assessments. If there are 1,718 researchers across 50 economics faculties, averaging 34 per faculty with 21% women, there would be an average of 7 women per faculty. However, universities may vary greatly in terms of overall numbers, and this distribution is important to see, especially when using average values in research. Let's suppose that, on average, there are 34 researchers per faculty; on average, 7 are women and 27 are men. The chance that among 27 men, there will be a researcher-«superstar» who will pull out the average value of bibliometric indicators for men of the whole faculty with his super-high indicators is higher than that such a «superstar» will be among 7 women. I'm not advocating for using median values instead, but readers must clearly comprehend how sensitive the faculty's average bibliometric indicators might be to adding or excluding a single observation, given the total number of observations. To be as transparent as possible we provided summary information on the size of economics and political science departments, and the number of men and women per department. Supplementary information including minimum and maximums are provided in Table A2.

Regarding the “superstar” effect, we conducted a thorough check for outliers, especially in smaller departments. These checks helped correct some measurement errors.

We write on page 8:

Before conducting the analysis by university departments, we performed an outlier check, which was particularly important given the small size of some of the departments. During this process, we identified some researchers with incorrect Google Scholar information, which we manually corrected to ensure the accuracy of our data.

Pertaining to the superstar effect, we also want to reiterate:

While we cannot entirely rule out the influence of the “superstar” effect, we believe it is not inconsistent with the critical mass argument. In fact, the critical mass framework could help explain how increased female representation in a department fosters an environment where women have greater opportunities to excel, with some potentially reaching “superstar” levels.

The article uses two groups of bibliometric indicators from Google Scholar—overall indicators and the last six years indicators. The overall indicators reflect a person's career over its entire course. The linkage between overall career achievements and the current department can be quite ambiguous. Perhaps a researcher secured a professor position, let’s say, at Harvard after holding positions at departments with different gender balances and garnered main career achievements outside of Harvard. It’s quite challenging to affirm that the gender balance in this 2023 on Harvard faculty correlates with all cumulative publication achievements of a person. The second group of indicators—the last six years—offers a more accurate timeframe. Therefore, the gender balance in 2023 might be more convincingly related to a researcher’s output in this term. Perhaps viewing bibliometric indicators over even shorter periods could allow for establishing a firmer affiliation with the faculty. It's also worth noting potential concerns about the robustness of the percentage of women on small faculties. The current measure of women’s share in 2023 can be sensitive to hiring one additional person at a small faculty, significantly altering the gender balance. This is a good point, which we can only address partially.

First, we cannot for whether someone has been at another university during her tenure. However, even if he or she does, she was probably an outperformer. Otherwise, she would not have moved to Harvard.

For the performance over the past two or three years, there is no bibliometric data readily available. We produce the charts for the bibliometric data in the past six years, the most recent information we have, and the results are virtually unchanged (see Figures A1 to A3).

Furthermore, we added a control variable for the percentage of assistant professors. We know that women’s representation should be higher among assistant professors, the category with potentially the least publications and citations. We also control for the proportion of faculty with the title of distinguished professor (see pages 13 and 14).

On page 12, it's stated: “...we manually verified the gender of all faculty members in Asian universities.”

---

## [Decision Letter · Decision Letter 1]

6 Mar 2025

PONE-D-24-39523R1Closing the publishing gender gap in economics and political science Does a critical mass matter?*PLOS ONE

Dear Dr. Stockemer,

Thank you for submitting your manuscript to PLOS ONE. After careful consideration, we feel that it has merit but does not fully meet PLOS ONE’s publication criteria as it currently stands. Therefore, we invite you to submit a revised version of the manuscript that addresses the points raised during the review process.

**Reviewer 1 raised several questions that I would like you to answer or comment on.**

**I also have a feeling that one of your descriptions of Google Scholar may not be entirely accurate (namely "Once scholars have created their Google Scholar page, all published content by said author is automatically added to her/ his website"). As far as I know, Google Scholar adds articles to the profile when the "add articles automatically" option is selected. I did not find any other information about this in the references you cited. Could you please check and confirm this?**

We look forward to receiving your revised manuscript.

Kind regards,

Andrey Lovakov, Ph.D.

Academic Editor

PLOS ONE

Reviewers' comments:

Reviewer's Responses to Questions

**Comments to the Author**

1. If the authors have adequately addressed your comments raised in a previous round of review and you feel that this manuscript is now acceptable for publication, you may indicate that here to bypass the “Comments to the Author” section, enter your conflict of interest statement in the “Confidential to Editor” section, and submit your "Accept" recommendation.

Reviewer #1: (No Response)

Reviewer #3: All comments have been addressed

2. Is the manuscript technically sound, and do the data support the conclusions?

Reviewer #1: (No Response)

Reviewer #3: (No Response)

3. Has the statistical analysis been performed appropriately and rigorously? 

Reviewer #1: (No Response)

Reviewer #3: (No Response)

4. Have the authors made all data underlying the findings in their manuscript fully available?

Reviewer #1: (No Response)

Reviewer #3: (No Response)

5. Is the manuscript presented in an intelligible fashion and written in standard English?

Reviewer #1:** ** (No Response)

Reviewer #3: (No Response)

6. Review Comments to the Author

**Reviewer #1:**  (No Response)

**Reviewer #3:**  (No Response)

7. PLOS authors have the option to publish the peer review history of their article (what does this mean? ). If published, this will include your full peer review and any attached files.

**Do you want your identity to be public for this peer review?** For information about this choice, including consent withdrawal, please see our Privacy Policy .

Reviewer #1: No

Reviewer #3: No

---

## [Editor Report · Decision Letter 2]

8 Apr 2025

Closing the publishing gender gap in economics and political science Does a critical mass matter?*

PONE-D-24-39523R2

Dear Dr. Stockemer,

We’re pleased to inform you that your manuscript has been judged scientifically suitable for publication and will be formally accepted for publication once it meets all outstanding technical requirements.

Kind regards,

Andrey Lovakov, Ph.D.

Academic Editor

PLOS ONE
---

## [Editor Report · Acceptance letter]

PONE-D-24-39523R2

PLOS ONE

Dear Dr. Stockemer,

I'm pleased to inform you that your manuscript has been deemed suitable for publication in PLOS ONE. Congratulations! Your manuscript is now being handed over to our production team.

Kind regards,

on behalf of

Dr. Andrey Lovakov

Academic Editor

PLOS ONE